# Perimount MAGNA Ease vs. INSPIRIS Resilia Valve: A PS-Matched Analysis of the Hemodynamic Performances in Patients below 70 Years of Age

**DOI:** 10.3390/jcm12052077

**Published:** 2023-03-06

**Authors:** Alessandra Francica, Filippo Tonelli, Cecilia Rossetti, Antonella Galeone, Fabiola Perrone, Giovanni Battista Luciani, Francesco Onorati

**Affiliations:** Department of Surgery, Dentistry, Paediatrics, and Gynaecology, Division of Cardiac Surgery, University of Verona Medical School, 37129 Verona, Italy

**Keywords:** aortic valve replacement, pericardial bioprostheses, hemodynamic valve performance

## Abstract

Background: During the past decade, the Perimount Magna Ease (PME) bioprosthesis has been implanted worldwide for aortic valve replacement (AVR). Recently, the INSPIRIS Resilia (IR) valve has been introduced as the newest generation of pericardial bioprostheses. However, few data have been reported about patients ≤70 years, and no comparisons in terms of hemodynamic performance between these two bioprostheses have been ever reported. Methods: Patients aged <70 years undergoing AVR were considered for comparison between PME (*n* = 238) and IR (*n* = 192). Propensity score (PS) matching was performed by logistic regression with adjustment for eight key baseline variables. The two prostheses were compared in terms of hemodynamic performances up to 3 years postoperatively. Sub-analysis according to prosthetic size-category was accomplished. Results: A total of 122 pairs with similar baseline characteristics were obtained from the PS-matching. The two prostheses showed comparable hemodynamic performances at one year (Gmean: 11.3 ± 3.5 mmHg vs. 11.9 ± 5.4 mmHg; *p* = 0.8) and at 3 years postoperatively (Gmean: 12.2 ± 7.9 mmHg vs. 12.8 ± 5.2 mmHg for; *p* = 0.3). The sub-analysis of size-category confirmed no statistical differences concerning the hemodynamic performances for each annulus size. Conclusions: This first PS-matched analysis demonstrated that the newly developed IR valve achieves the same safety and efficacy of the PME valve during mid-term follow-up in patients aged <70 years.

## 1. Introduction

The Carpentier–Edwards Perimount series of bovine pericardial valves were originally introduced in 1981 and have been continually improved since then [1,2,3]. Perimount Magna Ease (PME) represents the third generation of the Carpentier–Edwards portfolio and it has been implanted worldwide during the last decade. Excellent clinical and hemodynamic outcomes have been reported at long-term follow-up, even for patients aged <65 years [4,5,6]. Nonetheless, the use of biological valves in young or relatively young patients is still questioned by some authors [7,8]. Recently, INSPIRIS Resilia (IR) has been introduced as the latest generation of bovine pericardial valves. Its tissue aims to improve the durability of the valve, and reduced calcification over the time has been reported in vitro and in experimental animal models [9,10]. To date, its safety and efficacy have been reported in clinical practice [11,12], but scarce data exist in patients ≤70 years of age [12,13]. Of note, no comparisons between PME and IR in terms of hemodynamic assessment have ever been reported. Therefore, it is the aim of this study to compare the hemodynamic performances of PME versus IR prostheses in patients aged <70 years undergoing aortic valve replacement (AVR).

## 2. Material and Methods

### 2.1. Study Design and Population

From January 2010 to December 2012, a total of 689 consecutive patients underwent isolated or combined AVR with PME at the Division of Cardiac Surgery, University of Verona. More recently, from September 2017 to January 2022, 220 consecutive patients underwent isolated or combined AVR with the newly developed IR valve. For the purpose of the study, only patients aged >18 and <70 years were considered. All patients underwent median sternotomy, and the valve implantation technique has remained the same over the years for both prostheses. The valve is placed supra-annular using double-needled interrupted 2-0 synthetic braided pledgeted. Both PME and IR valve have a low-profile which make ease the implantation. The main difference consists in the dry storage of the new Resilia tissue, which does not need to be washed before the utilization.

Emergent and salvage cases were excluded.

Therefore, the overall study population included 238 patients receiving PME prostheses and 192 receiving IR valves. Finally, a Propensity Score (PS) matching analysis was performed in order to account for the baseline differences between these two cohorts, thus obtaining two homogenous comparable groups (Figure 1).

### 2.2. Endpoints

The primary endpoint of this study was to compare the hemodynamic performances between the latest generation of IR prosthesis and PME valve at short and mid-term follow-up. Prosthetic hemodynamic performance was evaluated by collecting echocardiographic data at one year and three years after surgery. According to the latest 2016 “Recommendations for the imaging assessment of prosthetic heart valves” from the European Association of Cardiovascular Imaging [14], the following parameters were assessed: trans-prosthetic mean gradient (Gmean), trans-prosthetic peak gradient; trans-prosthetic peak velocity (Vmax); presence, location (intra- vs. para-valvular) and severity of prosthetic heart valve regurgitation; severity of prosthetic stenosis; left ventricular end-diastolic volume (LVEDV); left ventricular end-diastolic diameters (LVEDD); thickness of interventricular septum (IVP); severity of mitral and tricuspid valve regurgitation; and systolic pulmonary arterial pressure (PAPs).

Secondary endpoint was to compare trans-prosthetic gradients and peak velocities stratifying by prosthetic size category (small: 19–21 mm; medium: 23–25 mm; large: 27–29 mm).

Postoperative outcomes were similarly secondary endpoints and were defined according to the latest “Valve Academic Research Consortium 3” (VARC-3) criteria [15].

### 2.3. Data Management

Pre-operative, intra-operative, and post-operative data were collected in a dedicated and anonymized database. Clinical follow-up was accomplished either by querying the Electronic Clinical Chart of each patient (retrieving data from the Regional Health Database) and/or by interviewing the patient. All echocardiographic data were retrieved from the Institutional Database of Cardiological Referring Hospitals, which were all accredited to European standards of Echocardiography [14]. Datasets were verified by a professional statistician who also verified homogeneity of data; similarly, a professional statistician carried out all the statistical analysis.

### 2.4. Statistical Analysis

Descriptive statistics were used to analyze data. Categorical variables are presented as absolute values and frequencies (%) and continuous variables as means with standard deviations (SDs) or median and interquartile range (IQR). All descriptive statistics are based on available cases. Group comparisons were carried out using *t*-test or Mann–Whitney U-test for continuous variables as appropriate depending on distribution, and a Fisher’s exact test for categorical variables. A Kolmogorov–Smirnov test was used to test continuous variables for normally distribution. PS-matching was performed to account for differences in patient characteristics at baseline. The propensity score for each patient was calculated by logistic regression with adjustment for eight key baseline variables, including age, diabetes, hypertension, dyslipidemia, peripheral vascular disease, prior myocardial infarction, bicuspid valve, and moderate/severe aortic regurgitation. A difference in propensity score of 9% (0.09) was tolerated when matching patients 1:1. As a measure of effect size, we provide Cohen’s d for the significant continuous variables. The statistical analysis was performed using SPSS Version 24.0 (IBM Corp. Armonk, NY, USA). A *p*-value of <0.05 was considered significant.

## 3. Results

### 3.1. Overall Population

The overall population included a total of 430 patients undergoing isolated (50.5%) or combined (49.5%) AVR: 238 and 192 receiving PME and IR prostheses, respectively. The patients belonging to the PME cohort were older (59.4 ± 10.0 vs. 56.3 ± 8.7 years; *p* < 0.01) and more likely to be affected by hypertension, diabetes mellitus, dyslipidemia, and peripheral vascular disease compared to IR group (*p* < 0.01) (Appendix A). On the other hand, patients who received IR valves showed higher incidence of bicuspid aortic valve disease (60.2% vs. 24.1% in IR and PME group, respectively; *p* < 0.01) and moderate/severe aortic regurgitation at the preoperative echocardiographic evaluation (Appendix A). No periprocedural deaths were reported; major outcomes were good and comparable among the two population (Appendix A).

### 3.2. Propensity-Matched Population

Once the PS-matching analysis was performed, 122 pairs of patients with similar baseline characteristics were analyzed (Table 1). The two populations had similar mean age (57.7 ± 11.1 and 57.0 ± 9.1 in PME and IR cohort, respectively; *p* = 0.09) and risk profile (EuroScore-II 2.7 ± 2.4%). Females represented <30% of both populations. Hypertension was the most common comorbidity (>50%) and about 30% of patients were scheduled in class NYHA II/IV in both groups. Isolated AVR was the most performed intervention (45.1% and 46.7% in PME and IR cohort, respectively; *p* = 0.79). The average clamp time was 88.5 ± 33.6 min and 83.9 ± 31.1 min for the PME and the IR implantation, respectively (*p* = 0.33) (Appendix A). The postoperative outcomes were still comparable between the two PS-matched populations, reporting an overall low incidence of major complications (Appendix A). Among the latter, paroxysmal atrial fibrillation was the most frequent complication (22% vs. 24% in PME vs. IR, *p* = 0.8). Postoperative type 1 stroke occurred rarely in both cohorts (0.8%), while a definitive PM was implanted in 3.3% and 1.6% of PME and IR patients, respectively (*p* = 0.4). All post-operative results are displayed in Appendix A.

### 3.3. Prosthetic Hemodynamic Performance Evaluation

At 1 year of follow-up, no moderate/severe prosthetic stenosis or regurgitation were assessed in both populations, as well as no paravalvular leaks (Table 2). The two prostheses showed a comparable hemodynamic behavior (Gmean: 11.3 ± 3.5 mmHg vs. 11.9 ± 5.4 mmHg for PME vs. IR valve; *p* = 0.8; Vmax: 2.20 ± 0.39 cm/sec vs. 2.26 ± 0.4 cm/sec for PME vs. IR valve; *p* = 0.3), as well as comparable left ventricular volumes and diameters (LVEDV: 120.3 ± 41.7 mL vs. 105.5 ± 29.8 mL for PME vs. IR group; *p* = 0.09; LVEDD: 51.3 ± 6.8 mm vs. 49.2 ± 8.1 mm, for PME vs. IR group; *p* = 0.08). However, the IVP thickness resulted lower in IR cohort (13.2 ± 2.1 mm vs. 12.1 ± 2.0 mm in PME vs. IR group; *p* < 0.01), as well as the PAPs (34.3 ± 9.3 mmHg vs. 26.8 ± 7.6 mmHg in PME vs. IR group; *p* < 0.01). Moreover, IR patients showed a lower rate of moderate/severe mitral regurgitation than PME patients (0% vs. 4.8%, respectively; *p* < 0.01) (Table 2).

At the 3-year echocardiographic evaluation, the PME cohort showed a higher rate of moderate/severe stenosis and regurgitation than IR prostheses, although not statistically significant (2.3% vs. 0%, respectively; *p* > 0.05). Severe SVD occurred only in four patients with PME. All of them were under 50 years old and underwent surgical reoperation. The prosthetic hemodynamic measurements remained comparable between the two groups (Gmean: 12.2 ± 7.9 mmHg vs. 12.8 ± 5.2 mmHg for PME vs. IR valve; *p* = 0.3) as well as left ventricular volumes and diameters (LVEDV: 116.1 ± 36.4 mL vs. 103.0 ± 32.1 mL for PME vs. IR group; *p* = 0.1; LVEDD: 50.5 ± 6.3 mm vs. 49.1 ± 3.4 mm, for PME vs. IR group; *p* = 0.5). All data related to hospital outcome are displayed in Table 2.

Sub-analysis related to the size-category reported no statistical differences between the PME and IR valves in relation to the Gmean (19–21 mm: 13.1 ± 3.1 mmHg vs. 15.4 ± 6.4 mmHg, *p* = 0.2; 23–25 mm: 11.6 ± 3.6 mmHg vs. 10.8 ± 4.4 mmHg, *p* = 0.08; 27–29 mm: 9.0 ± 2.2 mmHg vs. 9.31 ± 2.70 mmHg; *p* = 0.7, for PME and IR respectively) or Vmax (19–21 mm: 2.4 ± 0.32 mmHg vs. 2.58 ± 0.43 mmHg, *p* = 0.1; 23–25 mm: 2.22 ± 0.42 mmHg vs. 2.17 ± 0.39 mmHg, *p* = 0.5; 27–29 mm:1.98 ± 0.27 mmHg vs. 1.97 ± 0.33 mmHg; *p* = 0.9, for PME and IR respectively), for any size-category (Figure 2).

## 4. Discussion

To the best of our knowledge, this study represents the first propensity matched analysis comparing the IR versus the PME prostheses in patients aged <70 years, the latter being considered the gold standard available biological valve from Edwards Lifesciences given the well-acquainted results at long-term and very long-term follow-up [2,6]. This analysis demonstrates that the latest IR generation and its predecessor, the PME valve, showed comparable hemodynamic performances at short to mid-term follow-up. These outcome data were confirmed also when stratifying for each size-category.

Given that PME prosthesis has been the most implanted bovine pericardial valve of the last decade, these results are of note, in our opinion, given that only scant data are currently available for the IR valve in human practice. Hemodynamic stability of PME has been widely assessed [4,5,6,16,17], as well as its superiority in terms of durability when compared with other pericardial valves [16,17,18]. The long-term outcomes have been recently published, showing excellent freedom from structural valve degeneration (SVD), even for patients aged <65 years (92.6% at 12 years of follow-up) [6].

The IR prosthesis preserves the same design of the PME valve, though the new technology behind the tissue preservation aims to reduce the calcification over the time, so as to improve the valve durability. Mid-term results seem to confirm these expectations, as demonstrated by the most recent literature [12,19]. Johnston et al. [19] reported a transvalvular mean gradient of 11.0 ± 5.6 mmHg and no evidence of unexpected thrombosis or SVD at 4 years of follow-up. Then, Bavaria et al. [12] confirmed the stability of trans-prosthetic gradients at 5 years (Gmean 11.5 ± 6.0 mmHg), with 97.8% and 96.3% of patients showing none or trace of paravalvular and transvalvular regurgitation. In line with these findings, our IR cohort showed no moderate/severe stenosis or regurgitation at short- to mid-term, and the mean trans-valvular gradients were 11.9 ± 5.4 mmHg and 12.8 ± 5.2 mmHg at 1 and 3 years after surgery, respectively. Similarly, our sub-analysis on different size-categories reveals that our results are in line with those of previous studies. The COMMENCE trial, indeed, showed that the small-sized IR prostheses reported a mean gradient of 15–20 mmHg [12,19]. Similarly, we found a mean gradient of 15.4 ± 6.4 mmHg for 19–21 mm sizes of IR prostheses. Moreover, after matching with the same sizes of PME valve, no statistical differences were proven in trans-prosthetic gradients or peak velocity. Of note, some authors have already analyzed the PME small sizes. Anselmi et al. [20] reported a mean gradient of 14.3 ± 5 mmHg for the 19–23 mm PME, with an actuarial freedom from SVD at 5 years of 99.1%. Similar findings were observed by Nielsen et al. [21], comparing PME and Mitroflow valves. Looking at the small aortic annuli (size 19–21 mm), freedom from SVD at 10 years resulted in 100% versus 96% for PME and Mitroflow valves, respectively. They also found, by multivariate analysis, that bioprosthetic size did not represent a risk factor for SVD, as also proven by other authors [6,22]. Accordingly, we found low trans-prosthetic mean gradients in small PME (19–21 mm Gmean: 13.1 ± 3.1 mmHg). Of note, we found that the number of implanted valve sizes was still significantly different among IR and PME, even after PS matching. A larger number of small sizes of IR valve were implanted when compared with small sizes of PME. Nonetheless, when we performed the size category analysis, no significative differences were found between IR and PME in terms of transvalvular mean gradients for each prosthetic size.

Finally, this study showed a steady left ventricular reverse remodeling over time in both populations. This issue has been rarely investigated in PME or IR valves, but several studies have already demonstrated that postoperative left ventricle remodeling is a predictor of long-better outcome [23,24]. Izumi et al. [23] found that left ventricular function and reverse remodeling assessed at 1 year after AVR are useful predictors of long-term outcome, in both aortic stenosis and regurgitation. Corti et al. [24] demonstrated how a postoperative reduction in the end-diastolic dimension >20% predicts a significantly better late survival. In our study, looking at the 1-year echocardiographic assessment, we found comparable reduction of LVEDV, in both PME (from 176.9 ± 64.8 mL to 120.3 ± 41.7 mL) and IR population (from 146.2 ± 53.3 mL to 105.5 ± 29.8 mL). These values proved stable also at mid-term follow-up. All these findings support the excellent outcome recently reported at long-term follow-up for PME valves, and might suggest even better long-term outcome in younger patients receiving IR prostheses. To date, however, the superiority of the IR over PME valve in terms of durability has been assessed solely in vitro simulations [10] and only future clinical investigations will assess the potential advantage of the IR valve.

### Limitations of the Study

The main limitation of the study relates to the retrospective nature of our investigation. However, these results come from a real-world and all-comers daily practice.

Another limitation stems from the lack of an echocardiographic standardized protocol examination: echocardiographic follow-up was performed by cardiologists of different referring hospitals, though all echocardiographic labs were EU certified and comply with current guidelines [12]. It should also be considered that the study was carried out during the pandemic, and data retrieval was challenging. For example, some patients were not allowed to access the facilities for elective follow-up.

Finally, since IR prostheses were introduced at our institution in 2017, the patients of the two groups were treated almost 10 years apart. Consequently, IR follow-up is shorter than that of PME, and we had a lower amount of echocardiographic data at 3 years of follow-up. However, to the best of our knowledge, this is the first analysis which compares PME and IR prostheses in terms of hemodynamic performances in a relatively young cohort of patients undergoing AVR.

## 5. Conclusions

This first propensity matched analysis demonstrated that the newly developed IR valve achieves the same safety and efficacy of the gold standard PME valve at mid-term follow-up in patients aged <70 years.

## Figures and Tables

**Figure 1 jcm-12-02077-f001:**
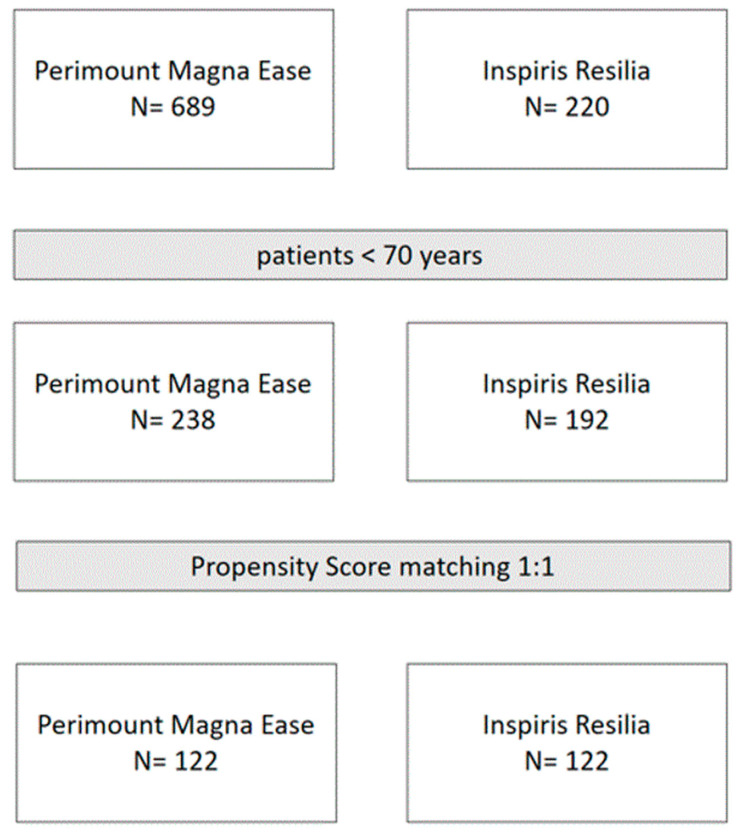
Consort diagram of study population.

**Figure 2 jcm-12-02077-f002:**
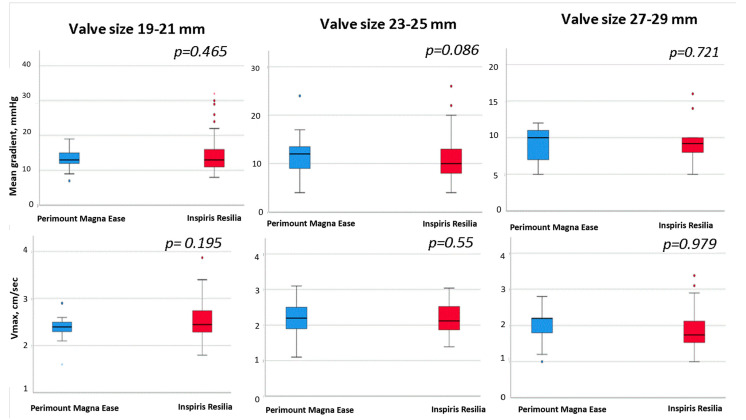
Perimount Magna Ease versus Inspiris Resilia prostheses: Gmean and Vmax compared by size-category (small: 19–21 mm; medium: 23–25 mm; large: 27–29 mm).

**Table 1 jcm-12-02077-t001:** Patient characteristics: Propensity Score matched population.

	Total	Perimount Magna Ease	Inspiris Resilia	*p*-Value Cohens d
	n/N (%) or Mean ± SD/Median (IQR) *n* = 244	n/N (%) or Mean ± SD/Median (IQR) *n* = 122	n/N (%) or Mean ± SD/Median (IQR) *n* = 122	
Age, years	57.4 ± 10.1 60 (52;65)	57.7 ± 11.1 61 (52;64)	57.0 ± 9.1 58 (52;64)	0.098
Female gender	55 (22.5)	24 (19.7)	31 (25.4)	0.284
BMI, kg/m^2^	26.9 ± 4.6	26.6 ± 4.2	27.1 ± 5.1	0.939
Diabetes mellitus	16 (6.6)	11 (9.0)	5 (4.1)	0.121
COPD	32 (13.1)	21 (17.2)	11 (9.0)	0.058
Creatinine, mg/dL	1.0 ± 0.8 238	1.1 ± 1.1 116	1.0 ± 0.4 122	0.291
Dialysis	4 (1.6)	4 (3.3)	0 (0)	0.122
NYHA				
I	51 (20.9)	4 (3.3)	47 (38.5)	<0.001
II	116 (47.5)	81 (66.4)	35 (28.7)	
III	40 (16.4)	10 (8.2)	30 (24.6)	
IV	37 (15.2)	27 (22.1)	10 (8.2)	
NYHA III/IV	77 (31.6)	37 (30.3)	40 (32.8)	0.679
EuroScore II, %	2.7 ± 2.4	2.7 ± 2.4	2.7 ± 2.4	0.788
Hypertension	133 (54.5)	68 (55.7)	65 (53.3)	0.700
Dyslipidemia	78 (32.0)	40 (32.8)	38 (31.1)	0.784
Active/former smoker	101 (41.4)	45 (36.9)	56 (45.9)	0.153
Peripheral vascular disease	28 (11.5)	18 (14.8)	10 (8.2)	0.159
History of stroke	9 (3.7)	6 (4.9)	3 (2.5)	0.500
Atrial fibrillation				
Paroxysmal	15 (6.1)	12 (9.8)	3 (2.5)	0.004
Persistent	30 (12.3)	20 (16.4)	10 (8.2)	
Pacemaker	4 (1.6)	3 (2.5)	1 (0.8)	0.622
Prev. MI	15 (6.1)	9 (7.4)	6 (4.9)	0.424
Prior PCI	15 (6.1)	8 (6.6)	7 (5.7)	0.790
Prior cardiac surgery	25 (10.2)	16 (13.1)	9 (7.4)	0.139
Coronary artery disease	47 (19.3)	22 (18.0)	25 (20.5)	0.626
Endocarditis	20 (8.2)	10 (8.2)	10 (8.2)	1.000
Ejection fraction, %	56.8 ± 10.4 60 (55;64)	57.1 ± 10.0 58.6 (55;65)	56.4 ± 10.9 60 (54;63)	0.932
**Preoperative Echo**				
Bicuspid valve	109 (44.7)	50 (41.0)	59 (48.4)	0.246
End-diastolic volume, mL	153.9 ± 57.7 140	176.9 ± 64.8 35	146.2 ± 53.3 105	0.012 −0.545
End-diastolic diameter, mm	59.8 ± 18.4 65	59.8 ± 10.2 36	59.8 ± 25.3 29	0.134
Interventricular septum, mm	13.0 ± 2.2 125	13.2 ± 2.2 50	12.8 ± 2.1 75	0.484
Peak gradient, mmHg	64.9 ± 26.7 99	62.8 ± 23.0 29	65.8 ± 28.2 70	0.623
Mean gradient, mmHg	43.3 ± 17.1 104	41.7 ± 17.3 29	43.9 ± 17.1 75	0.545
Vmax, cm/sec	3.8 ± 1.0 27	4.4 ± 0.9 2	3.7 ± 1.0 25	0.399
AVA, cm^2^	0.74 ± 0.36 67	0.46 ± 0.52 13	0.81 ± 0.27 54	0.165
Pulmonary artery pressure systolic, mmHg	38.6 ± 14.0 77	40.0 ± 11.6 22	38.0 ± 14.9 55	0.283
Aortic regurgitation				
No	70 (28.7)	36 (29.5)	34 (27.9)	0.174
Mild	47 (19.3)	18 (14.8)	29 (23.8)	
Moderate	40 (16.4)	18 (14.8)	22 (18.0)	
Severe	87 (35.7)	50 (41.0)	37 (30.3)	
Moderate/severe	127 (52.0)	68 (55.7)	59 (48.4)	0.249
Mitral regurgitation				
No	67/166 (40.4)	38/62 (61.3)	29/104 (27.9)	*<0.001*
Mild	71/166 (42.8)	14/62 (22.6)	57/104 (54.8)	
Moderate	20/166 (12.0)	8/62 (12.9)	12/104/(11.5)	
Severe	8/166 (4.8)	2/62 (3.2)	6/104 (5.8)	
Moderate/severe	28/166 (16.9)	10/62 (16.1)	18/104 (17.3)	0.844
Tricuspid regurgitation				
No	66/113 (58.4)	28/30 (93.3)	38/83 (45.8)	*<0.001*
Mild	37/113 (32.7)	2/30 (6.7)	35/83 (42.2)	
Moderate	6/113 (5.3)	0/30 (0)	6/83 (7.2)	
Severe	4/113 (3.5)	0/30 (0)	4/83 (4.8)	
Moderate/severe	10/113 (8.8)	0/30(0)	10/83 (12.0)	*0.046*

Legend: AVA, aortic valve area; BMI, body mass index; COPD, chronic obstructive pulmonary disease; MI, myocardial infarction; NYHA, New York Heart Association; PCI, percutaneous coronary intervention.

**Table 2 jcm-12-02077-t002:** PS matched cohort: one-year and 3-year echocardiographic data.

	Total	Perimount Magna Ease	Inspiris Resilia	*p*-Value Cohens d
	n/N (%) or Mean ± SD *n* = 244	n/N (%) or Mean ± SD *n* = 122	n/N (%) or Mean ± SD *n* = 122	
One-year echo				
Ejection fraction, %	57.2 ± 9.7 203	56.5 ± 9.7 86	57.6 ± 9.7 117	0.514
End-diastolic volume, mL	110.3 ± 34.6 121	120.3 ± 41.7 39	105.5 ± 29.8 82	0.091
End-diastolic diameter, mm	50.1 ± 7.6 78	51.3 ± 6.8 32	49.2 ± 8.1 46	0.081
Interventricular septum, mm	12.5 ± 2.1 123	13.2 ± 2.1 44	12.1 ± 2.0 79	0.011 −0.54
Peak gradient, mmHg	20.9 ± 7.9 203	20.2 ± 6.7 87	21.3 ± 8.7 116	0.624
Mean gradient, mmHg	11.7 ± 4.7 206	11.3 ± 3.5 87	11.9 ± 5.4 119	0.794
Vmax, cm/sec	2.23 ± 0.43 204	2.20 ± 0.39 87	2.26 ± 0.45 117	0.300
Pulmonary artery pressure systolic, mmHg	29.8 ± 9.0 63	34.3 ± 9.3 25	26.8 ± 7.6 38	0.001 −0.902
Prosthesis stenosis				
No	203/205 (99.0)	83/83 (100)	120/122 (98.4)	0.516
Mild	2/205 (1.0)	0/83 (0)	2/122 (1.6)	
Moderate	0/205 (0)	0/83 (0)	0/1222 (0)	
Severe	0/205 (0)	0/83 (0)	0/1222 (0)	
Moderate/severe	0/205 (0)	0/83 (0)	0/1222 (0)	n.a.
Prosthesis regurgitation				
No	199/206 (96.6)	80/84 (95.2)	119/122 (97.5)	0.447
Mild	7/206 (3.4)	4/84 (4.8)	3/122 (2.5)	
Moderate	0/206 (0)	0/84 (0)	0/122 (0)	
Severe	0/206 (0)	0/84 (0)	0/122 (0)	
Moderate/severe	0/206 (0)	0/84 (0)	0/122 (0)	n.a.
Paravalvular leak				
No	205/208 (98.6)	84/86 (97.7)	121/122 (99.2)	0.571
Mild	3/208 (1.4)	2/86 (2.3)	1/122 (0.8)	
Moderate	0/208 (0)	0/86 (0)	0/122 (0)	
Severe	0/208 (0)	0/86 (0)	0/122 (0)	
Moderate/severe	0/208 (0)	0/86 (0)	0/122 (0)	n.a.
Mitral regurgitation				
No	126/203 (62.1)	40/83 (48.2)	86/120 (71.7)	<0.001
Mild	73/203 (36.0)	39/83 (47.0)	34/120 (28.3)	
Moderate	4/203 (2.0)	4/83 (4.8)	0/120 (0)	
Severe	0/203 (0)	0/83 (0)	0/120 (0)	
Moderate/severe	4/203 (2.0)	4/83 (4.8)	0/120 (0)	0.027
Tricuspid regurgitation				
No	99/163 (60.7)	52/80 (65.0)	47/83 (56.6)	0.664
Mild	61/163 (37.4)	27/80 (33.8)	34/83 (41.0)	
Moderate	2/163 (1.2)	1(80 (1.3)	1/83 (1.2)	
Severe	1/163 (0.6)	0/80 (0)	1/83 (1.2)	
Moderate/severe	3/163 (1.8)	1/80 (1.3)	2/83 (2.4)	1.000
3-years echo				
Ejection fraction, %	58.7 ± 8.8 113	58.1 ± 9.5 88	61.1 ± 5.1 25	0.166
End-diastolic volume, mL	112.9 ± 35.6 108.5 (86.0;136.0) 68	116.1 ± 36.4 112.0 (87.0;137.0) 51	103.0 ± 32.1 100.0 (79.0;117.75) 17	0.189
End-diastolic diameter, mm	50.3 ± 5.9 57	50.5 ± 6.3 49	49.1 ± 3.4 8	0.552
Interventricular septum, mm	12.1 ± 2.2 73	12.3 ± 2.2 58	11.1 ± 1.8 15	*0.050*
Peak gradient, mmHg	22.8 ± 13.2 109	22.8 ± 14.2 84	22.7 ± 9.1 25	0.762
Mean gradient, mmHg	12.5 ± 8.2 110	12.5 ± 8.8 84	12.6 ± 5.5 26	0.680
Vmax, cm/sec	2.28 ± 0.55 109	2.26 ± 0.57 84	2.33 ± 0.46 25	0.621
Pulmonary artery pressure systolic, mmHg	31.0 ± 7.1 50	31.7 ± 7.9 37	28.9 ± 3.8 13	0.227
Prosthesis stenosis				
No	105/114 (92.1)	81/88 (92.0)	24/26 (92.3)	0.799
Mild	7/114 (6.1)	5/88 (5.7)	2/26 (7.7)	
Moderate	0/114 ((0)	0/88 (0)	0/26 (0)	
Severe	2/114 (1.8)	2/88 (2.3)	0/26 (0)	
Moderate/severe	2/114 (1.8)	2/88 (2.3)	0/26 (0)	1.000
Prosthesis regurgitation				
No	106/114 (93.0)	80/88 (90.9)	26/26 (100)	0.604
Mild	6/114 (5.3)	6/88 (6.8)	0/26 (0)	
Moderate	1/114 (0.9)	1/88 (1.1)	0/26 (0)	
Severe	1/114 (0.9)	1/88 (1.1)	0/26 (0)	
Moderate/severe	2/114 (1.8)	2/88 (2.3)	0/26 (0)	1.000
Paravalvular leak				
No	110/114 (96.5)	84/88 (95.5)	26/26 (100)	0.572
Mild	4/114 (3.5)	4/88 (4.5)	0/26 (0)	
Moderate	0/114 (0)	0/88 (0)	0/26 (0)	
Severe	0/114 (0)	0/88 (0)	0/26 (0)	
Moderate/severe	0/114 (0)	0/88 (0)	0/26 (0)	n.a.
Mitral regurgitation				
No	45/114 (39.5)	29/88 (33.0)	16/26 (61.5)	*0.016*
Mild	59/114 (51.8)	49/88 (55.7)	10/26 (38.5)	
Moderate	10/114 (8.8)	10/88 (11.4)	0/26 (0)	
Severe	0/114 (0)	0/88 (0)	0/26 (0)	
Moderate/severe	10/114 (8.8)	10/88 (11.4)	0/26 (0)	0.113
Tricuspid regurgitation				
No	55/101 (54.5)	47/85 (55.3)	8/16 (50.0)	0.848
Mild	44/101 (43.6)	36/85 (42.4)	8/16 (50.0)	
Moderate	2/101 (2.0)	2/85 (2.4)	0/16 (0)	
Severe	0/101 (0)	0/85 (0)	0/16 (0)	
Moderate/severe	2/101 (2.0)	2/85 (2.4)	0/16 (0)	1.000

## Data Availability

Data are contained within the article or Appendix A.

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
