# Peer review of "Perimount MAGNA Ease vs. INSPIRIS Resilia Valve: A PS-Matched Analysis of the Hemodynamic Performances in Patients below 70 Years of Age"

_jcm, 2023, doi:10.3390/jcm12052077_

Round 1
Reviewer 1 Report
The authors of this manuscript describe the results of their study comparing two biological valves. The study is of interest to readers and the results are clearly presented. Some limitations are not clearly described: The patients in both groups were treated almost 10 years apart, which may have an influence. There are some group differences, namely more bicuspid etiology which were corrected by propensity score matching. Implanted valve size was still significantly different after PSmatching. This should be discussed. If possible the authors should give more information on ease of implantation, surgical issues.
Reviewer 2 Report
Alessandra Francica et al present an interesting study on the hemodynamic performance of Perimount vs Resilia prosthesis.
I have anyway minor comments:
- Could the authors describe how many mitral valve replacement and repair they made? Do the authors think that the choice of a replacement/repair could impact on the hemodynamic performance of the prostheses?
- Were the patients with a degenerated bioprosthesis treated in any way? Or where they enlisted to perform a REDO/Valve in Valve re operation?
- I think that table 4 belongs to the main text. Nevertheless, I believe too many tables are in the supplementary data. The authors should try to make less table and to move them in the main text.
- Some typos (i.e. in the introduction section "Its tissue, aims...") and "about 30%" should be changed ("about" does not sound precise)
